# Thermal Degradation of Photoluminescence Poly(9,9-dioctylfluorene) Solvent-Tuned Aggregate Films

**DOI:** 10.3390/polym14081615

**Published:** 2022-04-15

**Authors:** Kang Wei Chew, Nor Azura Abdul Rahim, Pei Leng Teh, Nurfatin Syafiqah Abdul Hisam, Siti Salwa Alias

**Affiliations:** 1Faculty of Chemical Engineering Technology, Universiti Malaysia Perlis (UniMAP), Arau 02600, Perlis, Malaysia; chewkangwei@studentmail.unimap.edu.my (K.W.C.); plteh@unimap.edu.my (P.L.T.); nurfatinsyafiqah@studentmail.unimap.edu.my (N.S.A.H.); 2Advanced Optical Materials Research Group (AOMRG), Department of Physics, Faculty of Science, Universiti Teknologi Malaysia (UTM), Skudai 81310, Johor, Malaysia; siti.salwa@utm.my

**Keywords:** polyfluorene 1, ketonic defect 2, polyfluorene degradation 3, β-phase 4, photoluminescence polymer 5, conjugated polymer 6

## Abstract

The progression of the green emission spectrum during the decomposition of polyfluorenes (PFs) has impeded the development and commercialization of the materials. Herein, we constructed a solvent-tuned aggregated PFO film with the aim of retarding the material’s thermal degradation behavior which causes a significant decline in optical properties as a result of phase transformation. The tuning of the aggregate amount and distribution was executed by applying a poor alcohol-based solvent in chloroform. It emerges that at a lower boiling point methanol evaporates quickly, limiting the aggregate propagation in the film which gives rise to a more transparent film. Furthermore, because of the modulated β-phase conformation, the absorption spectra of PFO films were red-shifted and broadened. The increase in methanol percentage also led to a rise in β-phase percentage. As for the thermal degradation reactions, both pristine and aggregated PFO films exhibited apparent changes in the UV-Vis spectra and PL spectra. In addition, a 97:3 (chloroform:methanol) aggregated PFO film showed a more defined emission spectrum, which demonstrates that the existence of β-phase is able to suppress the unwanted green emission.

## 1. Introduction

Conjugated polymers are organic macromolecules characterised by an alternating backbone chain of double and single bonds. Their overlapping p-orbitals form a system of delocalised pi (π)-electrons, resulting in fascinating and valuable optical and electronic features [1]. These include the disclosure of electroluminescence polymers and their applications for optoelectronic devices, such as flexible solar panels for portable devices and organic light-emitting diodes in displays. Polyfluorenes (PFs) are one of the most convincing classical blue light-emitting materials suitable for polymer light-emitting diodes [2]. They have drawn much attention due to their high photoluminescence quantum yield, good charge-carrier mobility, and stable physical and chemical properties [3]. The two key advantages of PFs over other forms of light-emitting materials are extensive molecular engineering and ease of fabrication. Another advantage of PFs is that they can be readily dissolved in common organic solvents like chloroform, toluene, and tetrahydrofuran (THF), enabling the fabrication of large-area devices at a relatively low cost via spin coating and inkjet printing [4].

The polymorphism behaviour of poly(9,9-dioctylfluorene) (PFO) allows phase morphological variations during photophysic reaction without much chemical modification. In its solid state, PFO exhibits two main conformations. The first conformation is the glassy conformation and the second is the β-conformation. Because of PFO’s coplanar structure, β-conformation possesses a longer effective conjugated length than the glassy conformation, and this is a unique feature in contrast to other conjugated polymers [5]. Other benefits of having β-conformation include the improvement of charge-carrier mobility and efficiency in optoelectronic devices. Liang and coworkers demonstrated that increasing the phase content improves the performance and colour purity of polymer light-emitting devices (PLEDs), with the improved efficiency attributed to optimised photoluminescence and more balanced charge distribution in the light-emitting layer [6]. Although Zhang and colleagues also claimed that optimising β-phase formation in PFO films could result in improved PLED performance, with β-phase fractions of less than 5% exhibiting up to a 66.6% increase in photoluminescence quantum yield (PLQY) [7].

Nonetheless, PFs are prone to relatively fast degradation and suffer from broadband emission, which shifts the light wavelength from blue to green emissions [8]. The appearance of the green band will significantly affect the colour purity and decrease the device efficiency as well as stability. The green emission was initially explained by polymer aggregation and excimer formation, and later by keto defects. Most studies have suggested that fluorenone type defects can be formed thermally [9] and photochemically by direct excitation [10] or through single oxygen-mediated oxidation [11]. Typically, the green emission becomes more visible after annealing [12]. For instance, previous work by Gamerith et al. [13] reported that when exposed to excessive thermal stress, the presence of oxygen along with physical stress will cause PFs to emit a dominant PL at 530 nm. Zhao’s group also pointed out that thermal oxidation degradation and subsequent crosslinking of PFO contribute substantially to the intensification of green emissions [9].

Furthermore, low-energy emission bands are generally linked to emissive keto defect sites, interchain aggregation, and the existence of excimers formed during the degradation of the PFs. However, it is interesting that the dense, close-packing, and organised structure of β-phase, which is more stable, can suppress low-energy emitting species [14]. For example, Lupton and Becker [15] reported the reduced ability of β-phase to be photobleached on a single-molecule level. Chen et al. [16] also reported that forming β-phase in PFO can significantly reduce the emission of low-energy species. The theory behind this could result from the linear alkyl side chains of β-phase being adjacent to fluorene units, preventing neighbouring PFO main chains from approaching and producing green emissions. Another alternative explanation is that efficient Förster energy from the glassy phase to the β-phase efficiently prevents excimer formation [17].

So far, a wide variety of research, such as incorporating different functional groups, modifying side chains, and blending different kinds of polymers, has been conducted to establish their effectiveness towards thermal degradation. When regarding the β-phase in PFO, most researchers have concentrated on manipulating the β-phase conformation formation in films. Understanding the behaviours of different conformations in the film’s state is key to changing its condensed state structure and photophysical properties [18,19,20]. Numerous studies have also revealed the correlation between molecular conformation and various intrinsic polymer properties, such as the triplet and polaron formation rates and the optical gain. Nonetheless, the previous literature rarely reported the potential of varying proportions of β-phase in the film to suppress the green emission towards thermal degradation. The coexistence of diverse chain conformation in the film state and its impact on thermal degradation remains a poorly understood phenomenon that should be investigated further.

In this work, we developed an aggregate film by tuning up the amount of poor solvents added in PFO solution which was turned into a film via the free drop casting method. The generated film surface morphology, optical properties, and its ability to retard the formation of green emissions under thermal exposure were evaluated. The amount of poor solvent is an important parameter to induce the proportion of β-phase in the aggregated PFO films and the highly ordered β-phase conformation can be achieved by intermixing between good and poor solvents. The emergence of β-phase also brings several advantages including suppressing the degradation and green emission in the PFO films, leading to more stable optical properties under the thermal circumstances. 

## 2. Materials and Methods

### 2.1. Materials

PFO was obtained from Sigma-Aldrich (M) Sdn.Bhd (St. Louise, MO, USA) with molecular weight (M_w_) of above 20,000 and polydispersity index (PDI) of ~3.2. Spectroscopic-grade EMSURE^®^ brand (Merck KGaA, Darmstadt, Germany) chloroform was used as a good solvent to dissolve the PFO. Spectroscopic grade methanol, ethanol, and isopropanol were purchased from HmBG Chemicals (Hamburg, Germany). The alcohols were used as a poor solvent for intermixing with chloroform in the experimental works.

### 2.2. Preparation of Pristine PFO Films and Aggregated PFO Films

PFO films were prepared by the solvent casting technique by using chloroform as a good solvent. The drop cast approach was in accordance with previous studies [21]. The clear microscope glass slides with grade CAT.NO.1701 (ground edges, Labmart, South Plainfield, NJ, USA) and thickness of 1–1.2 mm were chosen as the substrate. Before deposition, the glass slides were ultrasonically cleaned by immersing them for 15-min intervals in distilled water, ethanol, and isopropanol and then rinsing them with distilled water before drying in the oven. Different concentrations of PFO solutions were prepared by diluting PFO in chloroform. The 0.001 M PFO solution was then deposited onto the glass slides and left to dry at room temperature to form a thin film. Meanwhile the aggregated PFO film was prepared by dissolving 0.001 M PFO into a good solvent of chloroform and a poor solvent of methanol, ethanol, and isopropanol respectively. The addition of a poor solvent aimed to induce β-phase formation. Various volume ratios of poor solvent to chloroform ranging from (0:100 to 10:90) were selected to establish aggregated PFO films with different structure. The mixture of solvent:non-solvent was stirred for 10 min with a magnetic stirrer to homogenise the mixture [22,23,24]. Subsequently, the solutions were drop casted onto the glass slides, and then left to dry at room temperature (26.7 °C). The UV-Vis reflectance spectra for the produced films are indicated in Appendix A in the supplementary materials section. 

### 2.3. Thermal Degradation Test

The thermal degradation test was conducted by exposing the samples in an oven (Memmert UN110). Both pristine PFO and aggregated PFO films were subjected to different temperatures and durations in the oven to justify their thermal degradation behaviour. The tests were conducted at the temperature of 60 °C and 90 °C for time periods of 1 h to time periods of 72 h.

### 2.4. Characterization

Ultraviolet-visible (UV-Vis) absorption measurements of different PFO films were carried out by the PerkinElmer UV/VIS/NIR 750 Lambda spectrometer (Waltham, MA, USA). β-phase fraction in the PFO films was calculated from the absorption spectra. The photoluminescence (PL) spectra of PFO films were obtained by using a PerkinElmer luminescence spectrophotometer Model LS 55 (Waltham, MA, USA). All the PL curves were normalised at 444 nm. The phase transition temperature of different PFO films was determined by differential scanning calorimetry (DSC), performed by TA Instruments DSC Q10 (New Castle, DE, USA). Scans were taken at a rate of 10 °C/min from 30 °C to 250 °C. However, the crystallinity of PFO films was analysed by utilising the Rigaku X-ray diffractometer, Model Smartlab 2018 (Tokyo, Japan). The measurements were recorded with a Cu Kα radiation at a scan rate of 0.1 s per step in the range of 2θ = 2° to 90°. ATR technique was performed to conduct the Fourier Transform Infrared (FTIR) analysis for the PFO films. Analysis was done by the PerkinElmer, Model L 128044 (Waltham, MA, USA) with a wave number range of 600–4000 cm^−1^, 16 scans and a resolution of 4 cm^−1^. Scanning electron microscopy (SEM) observations were performed with Hitachi TM3000 scanning electron microscope (Tokyo, Japan) to analyse the surface morphology of the PFO film. A voltage of 15 kV was used as the activation voltage to avoid the degradation of PFO films when running the test. Before testing, PFO films were coated with platinum by using the JFC-1600 Auto Fine Coater (JEOL Ltd., Tokyo, Japan). Luminescence spectra and decay time were recorded by using Jasco Photoluminescence Spectroscopy FP-8500. The sample was placed in the sample holder and was scanned for the radiation spectral wavelength of 350–700 nm with xenon lamp as the excitation source.

### 2.5. Calculation of β-Phase Fraction and Crystallinity Percentage and in PFO Films

It is known that the absorption spectra of mixed-phase PFO films are a linear superposition of glassy and β-phase contributions. The absorption spectra were normalised for the corresponding mixed-phase and referenced glassy absorption spectra first, followed by subtraction to determine β-phase fraction in the PFO films. β-phase fraction can thus be calculated by integrating the area in the range of 340–450 nm, of the difference absorption spectrum, Δ*A*, and the total absorption of the β-phase film sample, *A**_total_*, while accounting for the relative difference in oscillator strength *f_osc_* [25]. Earlier, a study by Huang et al. revealed that the glassy phase absorption is scaled by a factor of 1.08 to account for the difference in oscillator strength and β-phase chain segments [26]. As a result, β-phase fraction for the PFO films in this study was determined using the following equation:(1)$$\beta -\mathrm{phase}\;\mathrm{fraction}\;\left(\%\right)=\frac{\Delta A}{\Delta A+\left\{\;\left(A_{total}-\Delta A\right)\times 1.08\right\}}\times 100.$$

The crystallinity of all PFO films was calculated according to the following equation: (2)$$\mathrm{Crystallinity}\;\mathrm{percentage}\;\left(\%\right)=\frac{I^{’}}{I}\times 100$$
where *I*′ = the area of the intensity for the crystalline peaks and *I* = the area of the intensity of all peaks.

## 3. Results and Discussion

### 3.1. Production of PFO Aggregate Films

The type of poor solvent employed for aggregation purposes signifies a vital role in the architecture of the aggregate size, pattern, and distribution in polymer solution. Figure 1 illustrates the UV-Vis absorption spectra of pristine PFO film and aggregated PFO films induced by different ratios of chloroform to alcohols and the photoluminescence spectra of aggregated PFO films from chloroform/methanol mixture, respectively. Obviously, as the ratio of poor solvent increases, both the maximum absorption (λ_max_) and the absorption intensity for the films decrease simultaneously. The resemblance in terms of position and pattern suggests that the interaction between the binary solvent mixture and PFO impacts the photophysics of films [27]. As the methanol content increased, films cast from the mixture of chloroform/methanol showed a relatively higher absorption intensity. In comparison, the film cast from the mixture of chloroform/ethanol exhibited a significantly reduced absorption intensity after the ratio of 93:7. Whereas, it can be seen clearly in Figure 1b that the film cast with a 95:5 ratio of chloroform/isopropanol indicates a significant reduction in absorption peak. Thus, it can be inferred that the mixture of chloroform with a different type of alcohol would substantially affect the absorption intensity and the position of the spectral maxima of PFO films.

The merge ratio of a good to poor solvent plays a significant role in polymer aggregation in the solution state. Alcohol solvents are poor solvents for conjugated polymers; therefore, when PFO molecules are in close proximity to the alcohol solvent molecules, they induce strong aggregation effects. The polarity of alcohol decreases with increasing alkyl group length [28]. Hence, compared to chloroform/ethanol and chloroform/isopropanol, the synergistic solvation was found in the chloroform/methanol mixture, thus offering a more polar environment, allowing PFO to solvate more effectively [29]. However, during casting, film production is complicated by the interaction of solvent evaporation and molecule aggregation; one would assume that the solvent used to coat thin films will profoundly influence the morphology [30]. The higher boiling point solvent leads to a longer evaporation time, resulting in higher crystalline order in films and highly aggregated structures [19]. This is in agreement with the study by Kwon and coworkers (2020) [31].

In our work, the lower boiling point methanol evaporated quickly, thus limiting the propagation of the aggregates in the film. In comparison, the high boiling point isopropanol allowed it to persist longer than chloroform during evaporation, resulting in larger aggregation sizes and rougher morphology. Following rapid evaporation of chloroform, the remaining isopropanol exerted a greater influence on the crystallisation behaviour of PFO, and isopropanol’s poorer solvation ability promoted high aggregation [31].This phenomenon could be corresponded to the absorption spectra as shown in Figure 1. Usually, UV-Vis measurement is caused by the change of crystallinity and a consequent change in the light absorption of each film. As UV-Vis measurement requires higher transmittance from the film, larger aggregation sizes will cause the film to become opaque to UV radiation and reduce absorbance intensity.

Figure 1d depicts the PL spectra of the PFO film and aggregated PFO films. The pristine PFO film and chloroform/methanol aggregated films exhibited fairly similar characteristics at the absorption (0–0 transition), and the emission part at longer wavelengths (0–1 transition) are substantially different. In contrast to the PL spectrum of the PFO film, the 97:3 aggregated film was characterised by a slight red shift but higher PL intensity. The red shift is consistent with a longer effective conjugation length as reported previously [32]. A rather broad emission spectrum has also been noticed in 97:3 aggregated film, as the intermolecular distance varied due to disorder. These effects are frequently observed and are caused by the migration of exciton to the most organised and lowest energy-gap regions in the film [33]. From the result, it has been demonstrated that among different conformations of π-conjugated polymers in solid-state samples, even minor changes in the interchain distance result in shifts in the emission energy [34].

The UV-Vis absorbance spectra of pristine PFO film and aggregated PFO films are revealed in Figure 2a. Pristine PFO film prepared by chloroform solution consists of only one absorption peak at 381 nm, appointed to the inhomogeneous broadened S_0_→S_1_ 0–0 transition of glassy phase. By comparison, the spectra for aggregated films were red-shifted and slightly broader, with an apparent low-energy shoulder emerging at around 433 nm, characteristic of the S_0_→S_1_ 0–1 and 0–0 transition of typical β-phase PFO. Zhang et al. (2017) stated in their study that this is due to the transition of the inter-monomer torsional angle of the PFO from glassy conformation to the well-defined β-phase chain conformation [7]. The close proximity of π-electrons in the aggregated state permitted them to delocalize throughout several stacked segments of polymer chains in both electronic ground and excited states. The degree of π-electron delocalization is sufficient to generate the electronic species with a lower HUMO–LUMO energy gap, which result in the red shift absorption. Thus, it can be clarified that methanol (poor solvent) served as a driving force to shorten the distance between PFO molecules. As a result, PFO molecules have a better chance of forming β-phase van der Waals interactions between interdigitated side chains, which lead to the formation of the planar PFO main chains [35].

Furthermore, Figure 2b demonstrated the percentage of β-phase as a function of methanol content. As stated in the prior paragraph, the fraction of β-phase presented in aggregated films was calculated by using the integration method. β–phase fraction was found to have an upper limit of 15.9%. We can also note that with an increase in methanol percentage, the shoulder peak also continues to rise. The absorption spectra were found to undergo a slight red shift from 433 nm to 435 nm with the increase of methanol percentage. This may be attributed to π–π interactions leading to the aggregation effects [36]. As compared to the glassy chain ensemble, β-phase absorption spectra display a lower degree of inhomogenous broadening related to its well-defined planar-zigzag geometry. It is well known that the aggregation of intra- or inter-polymers through interactions often results in red-shifted emissions and a less-resolved vibronic structure. For chloroform/methanol ratio 91:9 and 90:10, there is a sharp rise in the peak of β-phase compared to others, and this phenomenon can be defined as macroscopic aggregates. Huang et al. (2012) stated that the notable red shift in absorption spectra for main bands of those two films is evidence of macroscopic aggregates [26].

### 3.2. Thermal Induced Degradation of Aggregated PFO Films

Figure 3 shows the absorption spectra of the pristine and aggregate films of PFO after being subjected to the thermal degradation test at different temperatures. It is noticed that temperature differences cause visible variations in spectra and the absolute magnitude of the main peak. At 60 °C, each film’s absorption spectrum still retained its shape. However, with increasing temperature to 90 °C, a blue shift from 382.6 nm to 381.2 nm is observed in absorption spectra of pristine PFO film. This phenomenon is ascribed to the decrease in the effective conjugation length of PFO. Besides, the spectra’s broadening was notified after the thermal degradation test. This points to the fact that several chemical defects are introduced during the thermal degradation test, causing the conjugated backbone to be broken into shorter conjugated segments.

Furthermore, increasing temperature also caused aggregates to dissociate within their system. The simultaneous variation in absorbance intensity implies that increasing temperature decreases in the population of chromophores with relatively long conjugation lengths in both their electronic ground and excited state. At 90 °C, the aggregate shoulder of 97:3 aggregated film diminished from 434.4 nm to 430.4 nm, whereas the absorption maximum peak experienced a slight shift from 383.4 nm to 382.60 nm in absorption spectra. The blue shift of absorption spectra indicates a substantial decrease in the effective conjugation length, which could be influenced by conformational changes resulting in a lesser planar conformation. The same phenomena were observed for the 91:9 aggregated film. The macroscopic aggregation in 91:9 film caused the morphology to be rougher after the thermal test and induced further lower absorption intensity; thus, the thermal degradation test further proceeded for pristine PFO film and 97:3 aggregated film.

To further clarify the effect of the thermal degradation test on the films, we conducted the test at 90 °C at different duration of time. Figure 4a,c depict the UV-Vis absorption spectra of pristine PFO and 97:3 aggregate films over time. After 72 h, the broad, structureless profiles with lower absorption intensity are observed for the pristine PFO film and 97:3 aggregated film. Aside from the decrease in conjugation length, it is reasonable to believe that the blue shift in absorption maximum occurred after the formation of a radical at the C-9 position. When the radical attacks the conjugated backbone, resulting in crosslinking of polymer chains and backbone distortion as reported by Grisorio et al. (2011) [10]. 

A similar trend is seen in Figure 4b, the PL spectrum of pristine PFO film. Pristine PFO film exhibited a broadening PL linewidth in the 0–1 transition, together with the undesired green emission tail. The broadening of the PL spectrum coincides with the polymer phase’s disorder. The unwanted green tail emission would be attributed to on-chain ketonic defects as reported previously [13,37]. As the thermal test duration increases, the low-energy emission band around 520 nm is more significant and intensifies smoothly. This could be because a rise in fluorenone concentration results in an augmentation of the low-energy emission band relative to the pure PFO emission, as energy is transferred from the higher band gap polymer to the lower band gap fluorenone [38]. The overall spectra demonstrate the blue shifts as the duration of time increases, but the relative intensities of the individual vibronic peaks also vary. However, a more defined emission spectrum, the line narrowing of the 0–0 and 0–1, was detected in the aggregated film. This is consistent with Chen’s study, which demonstrated that the existence of a more organised β-phase is sufficient to suppress the emission of low-energy emitting species [39]. Furthermore, exciton transfer rates are strongly reliant on the local microscopic geometry and conformation of the molecules [33]. Other conjugated polymers, such as poly(p-phenylene vinylene), have shown an analogous blue shift in PL energies with increasing temperatures, which has been linked to lattice fluctuations [40]. Unlike single crystals, π-conjugated polymers solid-state samples ensembles of sites and crystalline domains with varying energy. One probable explanation is that when the temperature rises, the excitons generated on the polymer backbone do not easily migrate to the lower energy segments, instead remaining localized on the shorter-chain segments with higher energies [40]. Nevertheless, green emissions from degraded PF would be induced not just by fluorenone, but also by nonemissive quenchers such as alkyl ketone, due to their ability to effectively quench bulk emission [38]. 

Thermal analysis is performed to determine the nature of the crystallisation and transition temperatures of PFO films. Figure 5 demonstrates the corresponding DSC heating traces for all films. All films show a clear endothermic peak around 154 °C, which correlates to the melting peak of PFO [41]. Kawana et al. (2002) also reported that PFO exhibited an endothermal melting peak at T_m_ ≈ 157 °C [42]. Additionally, crystallisation of PFO film is observed in cooling at around 120 °C. The DSC data is consistent with PFO transitions that have been previously observed [43]. The results also indicate a noticeable change with the presence of the endothermic peak at 143.6 °C before the film reached its melting temperature at 153.9 °C. This is because the aggregate film undergoes a phase transition and tends to reorganize into the more crystalline phase upon heating. The same finding was reported by Chen et al. (2005), which stated that the phase transformation of PFO in the film is most frequently observed around 140 °C upon heating [44]. As a result, it could be inferred that the β-phase is usually metastable, evolving into more crystalline phases before the final melting of crystalline order at approximately 160 °C. The aggregated film appears to be more stable and produces a less pronounced changed in T_m_ values. The observed, relatively stable T_m_ could be correlated to the more ordered film behaviour, as it can compensate for the change of T_m_ upon thermal treatment. The crystallinity of the films was ascertained by XRD, as depicted in Figure 6.

XRD scans showed the diffractions at ~6.7°, ~15.5°, and ~20.1° for the pristine PFO film. On the other hand, more apparent diffraction at ~7° emerges in the XRD patterns of the aggregated film. This diffraction angle corresponds to the β-phase of PFO and is consistent with the prior result [45]. After the thermal degradation test, it can be seen that the peak of the pristine PFO film at ~15.5° diminished and the peak at ~20.1 widened, indicating that the macromolecule’s lateral approach was varied. On heating, the broadened peaks result from the ordered–disordered transition. The loss of crystallinity at elevated temperatures indicated the formation of structural disorders [4]. Thus, it seems reasonable to consider that the change in T_m_ and the appearance of peaks at around 112 °C was the loss of crystallinity caused by the thermal degradation test. Besides, the XRD findings indicate that the aggregate films possess higher crystallinity than the pristine PFO films, this owing to the possibility of an extra transition between them. The extra transition was attributed to the slow evaporation rate of methanol, which caused the PFO backbones to adopt more planar and ordered conformation [34]. Thus, the relative intensity of the diffraction characteristics associated with the aggregated film increase. In contrast, the crystalline structure of the pristine PFO film is weaker, as indicated by the presence of fewer and broader reflections that pertained to the backbone arrangement of PFO.

Figure 7 illustrates the FTIR spectra of different types of films before and after the thermal degradation test. The observed bands can be categorised into three regions: the region around 3000 cm^−1^, which is attributed by C-H stretching vibrations; the region below 1500 cm^−1^, which is associated with C-H scissoring vibrations; and the region between 1600 cm^−1^ and 1800 cm^−1^, where the feature emerges substantially after 72-h thermal degradation tests. The band appears at around 1609 cm^−1^, could be interpreted as the stretching mode of an asymmetrically substituted benzene ring. However, the small peak that occurs at around 1715 cm^−1^ is consistent with the fluorenone building block’s carbonyl stretching mode. The emergence of the two IR peaks indicates oxidation to fluorenone-containing structures, as fluorenone itself has shown strong bands at 1708 cm^−1^ and 1600 cm^−1^. This phenomenon has been found previously for various PFs during thermal and photo-oxidative degradation [46]. However, in our study, the bands at around 1715 cm^−1^ were not sufficiently intense, implying that although the green emission was induced by thermal oxidation degradation (not only by fluorenone concentration), it could also be accompanied by the crosslinking in the PF chains. When comparing IR spectra, 97:3 aggregate films appear at a lower degradation dose and thus indicate a lesser loss in spectral intensity. Furthermore, the PL feature of aggregate films reveals a higher resistivity because it appears to provide more efficient energy migration, giving a significantly lower defect contribution. 

The morphology of the films was examined by scanning electron microscopy (SEM). Figure 8 demonstrates the images of pristine and aggregate PFO films before and after being subjected to the thermal degradation test at 90 °C. The pristine PFO film and 97:3 aggregated films exhibit smooth surfaces, whereas the 97:3 aggregated film displays small domains that are most certainly microscopic aggregates. The similar surface topography of β-phase film was previously reported by Chen et al. (2005) [44]. In contrast, the 91:9 aggregated film features a rougher surface with larger domains that correspond to the macroscopic aggregates. Here, it can be observed that the β-phase of PFO exists as a more crystalline phase, and the conformational transition from glassy state to the β-phase definitely affects the molecular packing of PFO [47,48]. After the thermal oxidation degradation test, there are observable differences in the morphologies of PFO films. As the pristine PFO film contains an adequate amount of free space for oxygen diffusion, this leads to local changes in chain conformation and packing density. As a result, the surface became more rough. Compared with pristine PFO film, 97:3 aggregated film demonstrated smoother morphology. This could be due to more planar and crystalline conformation and a slower oxygen diffusion into the film. 

PFs are known to degrade rapidly in ambient environments, leading to the emergence of a broad, featureless green emission band at approximately 530 nm. The origins of this lower energy emission have long been disputed. At the moment, it is commonly accepted that ketone defects are unambiguously relevant for green emissions [49]. Thermal degradation is a well-documented problem throughout the ageing process of polymeric materials [50]. Often, the process included thermal-induced breaking of chemical bonds in the main and side chains. Chain scission happens more rapidly in the presence of oxygen, leading to the formation of partially oxidized polymeric species such as fluorenone [9]. Concurrent to the study of green emission, the mechanism of PFs thermal oxidative degradation has been proposed in Figure 9 to account for the green emission caused by thermal oxidation and crosslinking. 

The oxidation kinetics is an autocatalytic radical-chain process that tends to form emissive fluorenone and nonemissive alkyl ketone. Under the severe conditions of the thermal oxidation test, the formation of radical species (a) initiates the mechanism that provokes the degradation of the alkyl segments of fluorene units. The radical species (a) then interacts with oxygen to form peroxy radicals (b). In the propagation process, peroxy radicals (b) abstract hydrogen from the polymer’s side chain, forming peroxide (c) and radicals (d) in the side chain. The radicals (d) in the side chain also can react with oxygen to abstract hydrogen from the polymer, resulting in the peroxide (f). The peroxide (c) and (f) undergo an analogous oxidation reaction to generate fluorenone and alkyl ketone, respectively. In addition, crosslinking is also detectable as a result of chain termination via a radical–radical combination.

## 4. Conclusions

The type of poor solvent chosen to induce aggregation plays an important role in determining the aggregation size in PFO films. The formation of β-phase was further evidenced by a red shift in the PL spectra which attributed to the formation of a longer conjugation length. It is noticed that as the methanol content increased, the fraction of β-phase chain in the aggregated films increased as well. After being subjected to the thermal degradation test, the broad, structureless profile absorption spectra were observed for both the pristine PFO film and aggregated PFO film. Undesired green emissions accompanied by the blue shift was noticed in PL spectra for pristine PFO film, which indicated the formation of fluorenone or backbone distortion. In contrast, a more defined emission spectrum was detected for the aggregated film. This suggested that a more organised β-phase could suppress the emission of low-energy species. DSC, XRD, FTIR, and SEM analyses were used to verify and corroborate the results. Our study demonstrates that the optimised β-phase proportion in the aggregated PFO film can suppress the green emission and mitigate the effect of thermal degradation. This motivates further exploration on the potential of β-phase conformation in PFs for suppressing the degradation issue in PFs.

## Figures and Tables

**Figure 1 polymers-14-01615-f001:**
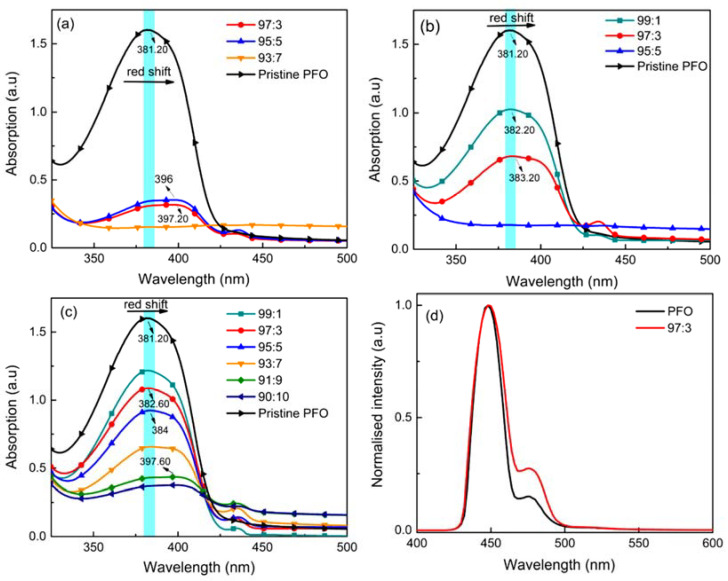
UV-Vis spectra of the PFO film and aggregated PFO films using various ratio of good to poor solvent of (**a**) ethanol, (**b**) isopropanol, and (**c**) methanol, and (**d**) normalised PL spectra for PFO and PFO with 97:3 induced aggregated films.

**Figure 2 polymers-14-01615-f002:**
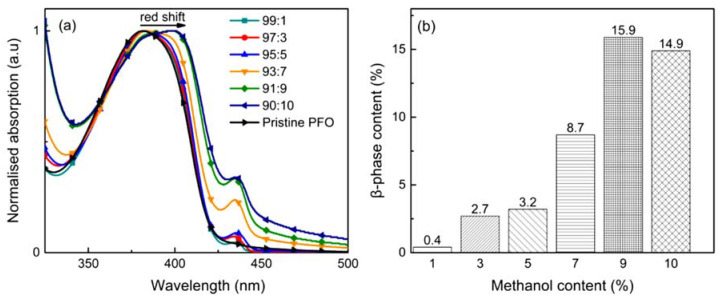
(**a**) UV-Vis absorption spectra of pristine PFO film and aggregated PFO films with different chloroform/methanol ratios and (**b**) the ratio of methanol content to the fraction of β-phase induced in aggregated films.

**Figure 3 polymers-14-01615-f003:**
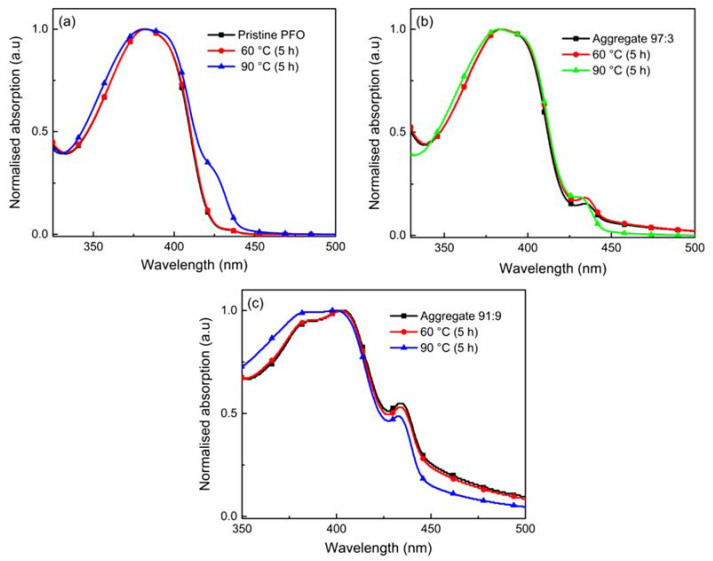
UV-Vis absorption spectra of (**a**) pristine PFO films and aggregated PFO films from chloroform/methanol with the ratio of (**b**) 97:3 and (**c**) 91:9, before and after the thermal degradation test.

**Figure 4 polymers-14-01615-f004:**
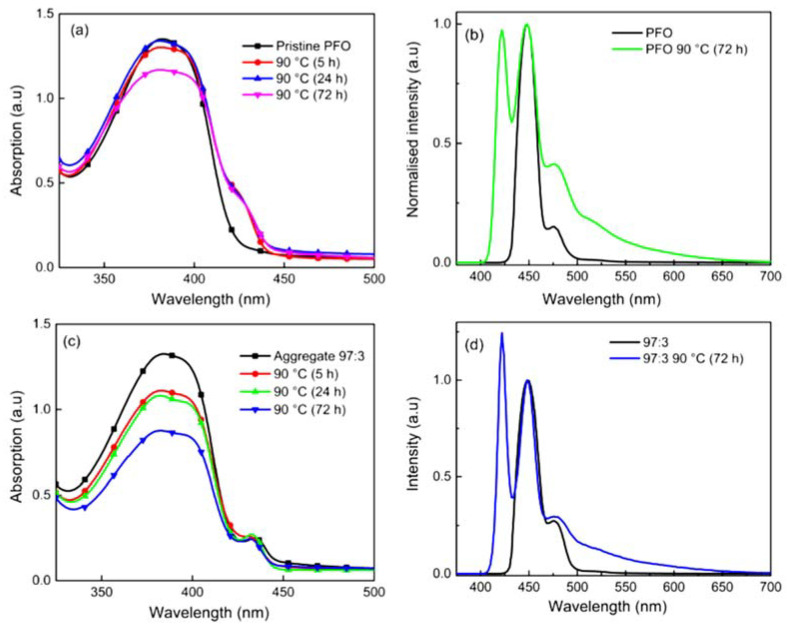
UV-Vis absorption spectra of (**a**) pristine PFO film and (**c**) 97:3 aggregated PFO film, and PL intensity spectra of (**b**) pristine PFO film and (**d**) 97:3 aggregated PFO film, before and after thermal degradation test.

**Figure 5 polymers-14-01615-f005:**
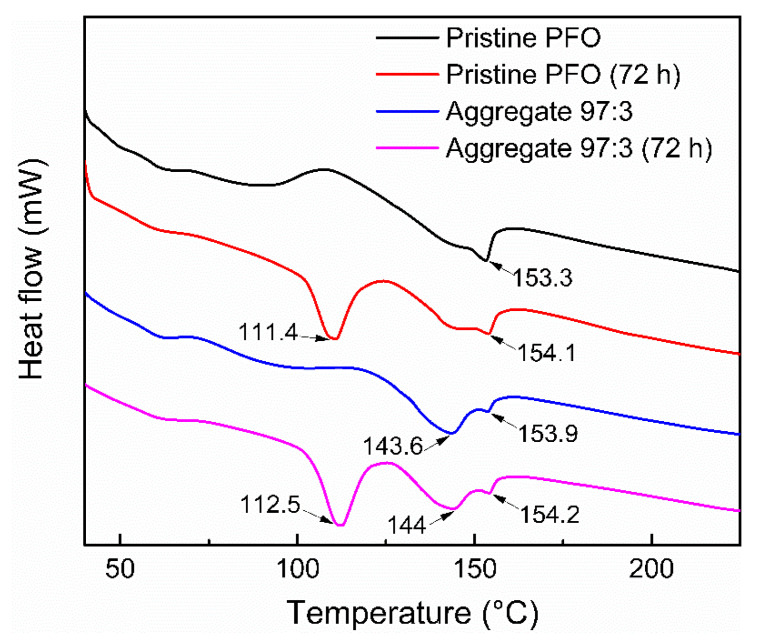
DSC heating curves of pristine PFO films and 97:3 aggregated PFO films before and after thermal degradation test at 90 °C for 72 h.

**Figure 6 polymers-14-01615-f006:**
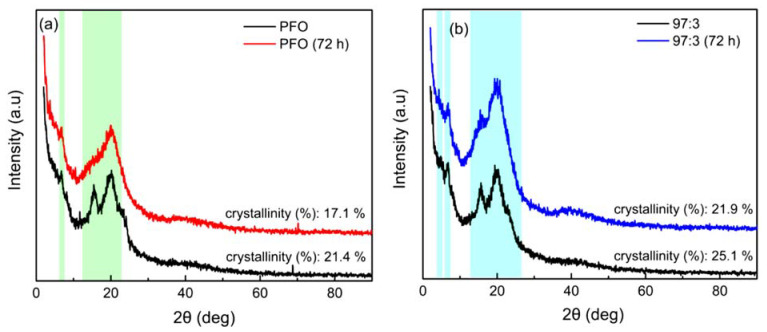
XRD analysis of (**a**) pristine PFO films and (**b**) 97:3 aggregated PFO films before and after thermal degradation test.

**Figure 7 polymers-14-01615-f007:**
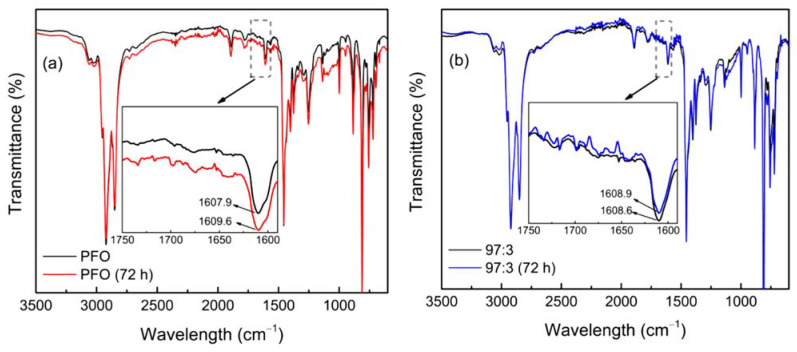
FTIR spectra of (**a**) pristine PFO films and (**b**) aggregated PFO films before and after thermal degradation test, respectively.

**Figure 8 polymers-14-01615-f008:**
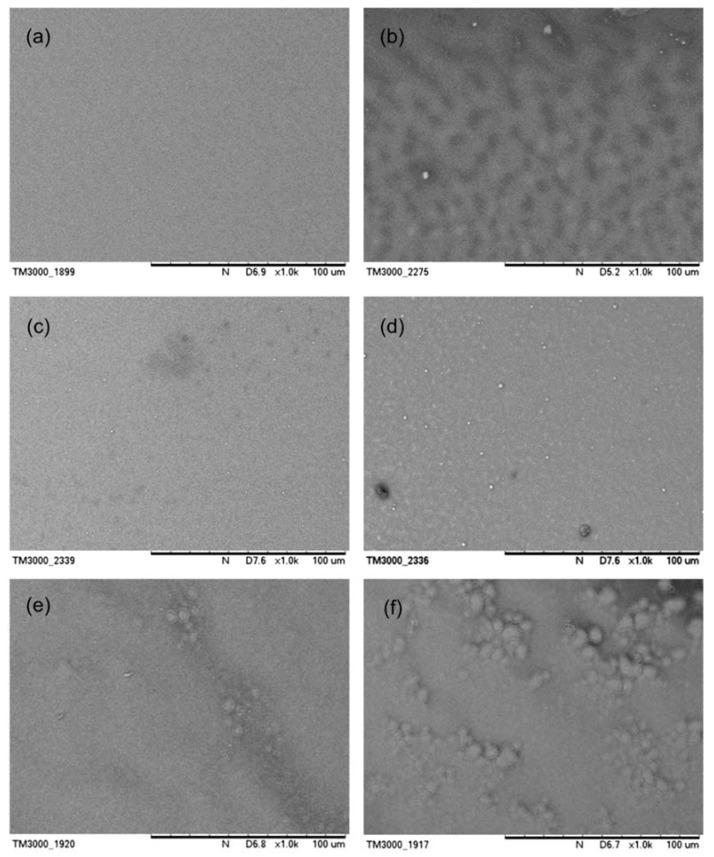
SEM images of (**a**) pristine PFO film, (**c**) 97:3 aggregated PFO film and (**e**) 91:9 aggregated PFO film, respectively, before the thermal degradation test. SEM images of (**b**) pristine PFO film, (**d**) 97:3 aggregated PFO film and (**f**) 91:9 aggregated PFO film, respectively, after thermal degradation test.

**Figure 9 polymers-14-01615-f009:**
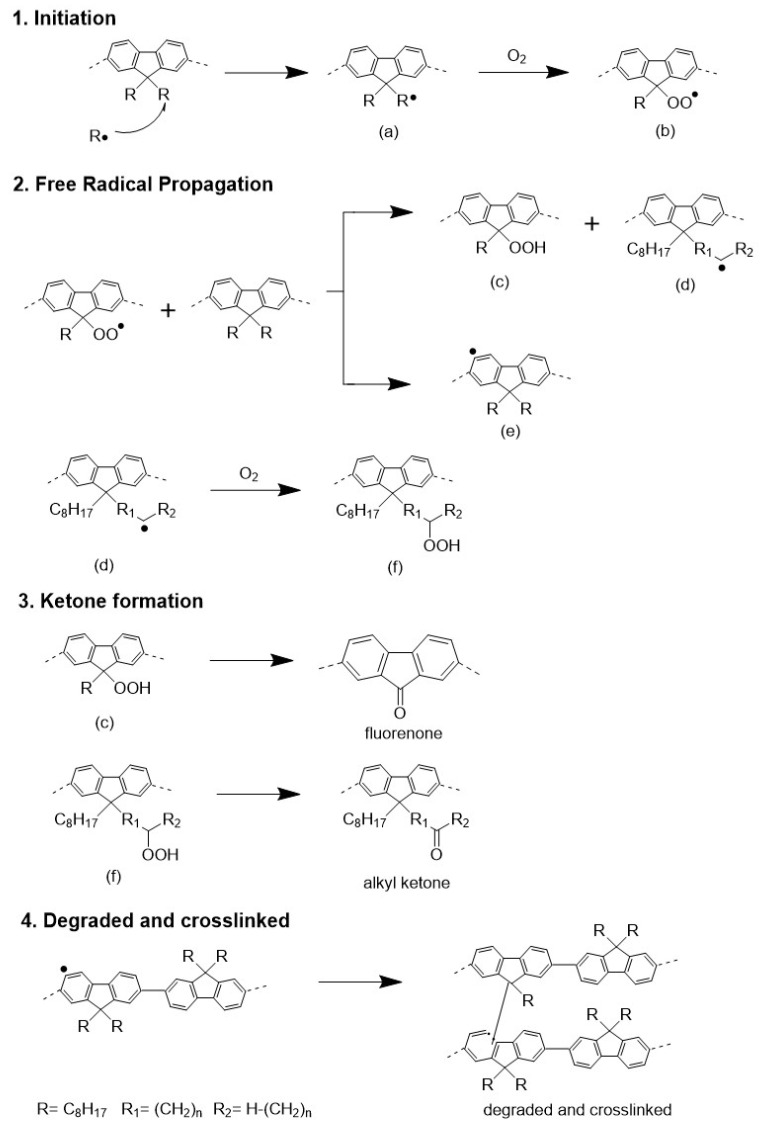
Thermal oxidation degradation pathway for aggregated PFO films.

## Data Availability

The data presented in this study are available in the supplementary materials.

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
