# Peer review of "Thermal Degradation of Photoluminescence Poly(9,9-dioctylfluorene) Solvent-Tuned Aggregate Films"

_polymers, 2022, doi:10.3390/polym14081615_

Round 1

Reviewer 1 Report

The paper polymers-1645366 deals with the effect of solvent-tuned aggregated of PFO films. The authors reports different experimental technique, mainly oriented on the optical properties of the materials. The discussion is clear and the reference are well balanced.

I have two concerns:

The first is related to the FTIR spectra reported figure 7. The authors at 1708 and about 1600 cm-1 (inset). The intensity of the first peak is very low and a discussion about it is not convincing with the reported experimental data. The authors should try to obtain a more significant spectrum and/or a softer assignment. Is the resolution of the instrument able to separate between 1608.9 and 1908.6 in a such large band?

The second point concerns on the PL intensity. You report only normalised spectra. Hence, it is not clear "the variation of the PL intensity" reported in the text (ie. line 208 and caption in figure 4). 

Author Response

Thank you for the valuable comments. Here, we attached the rebuttal for reviewer 1.

Reviewer 2 Report

In this paper, authors reported the Investigation of Thermal degradation of photoluminescence poly (9,9-di- 2 octylfluorene) solvent-tuned aggregate films. This work is interesting, but parts of the concept are not novel. However, it needs major revision before it can be accepted.  Some of the corrections and suggestions are as follows:

  1. I recommend rewriting introduction and exhibit your innovation.
  2. The authors did not describe carefully the choice of substrate and did not talk about the method to measure thickness of the layers and their values.
  3. Please define I¢ in the equation 2.
  4. Please add a vertical line in figure to show the reader the red-shifted.
  5. Line 231 ''As stated in the prior chapter'', please specify that section or paragaraph and not chapter.
  6. Line 294: please specify the full name of this PPV abbreviation.
  7. In figure 4 (b) and (d) please add °C and not C.
  8. Please add space between value and unit throughout text and between the end and the beginning of the next sentence (See 321 for example).
  9. You can superimpose the figures ((a) and (b)) and ((c) and (d)) in figure 5.
  10. The authors should add new references in the manuscript.

Author Response

Thank you for the valuable comments. Here, we attached the rebuttal for reviewer 2.

Reviewer 3 Report

Poly(9,9-dioctylfluorene) (PFO) is a widely studied blue-emitting conjugated polymer. Its optoelectronic properties are strongly affected by the presence of a well defined chain-extended “b-phase” conformational isomer. In this study efforts are presented to create β-phase formation in PFO films by using a good-bad solvent mixture approach. The manuscript is quite well written and the authors have tried to understand the effect of the ratios of solvent: non-solvent mixtures to the optical and structural properties of PFO.

One important fact that definitely needs clarification is the examined films thicknesses since this is crucial for the intensity of the absorption and emission profiles of the films and the whole behavior of the material examined. 

Several other points that need clarification and must be answered before the manuscript could be reconsidered for publication in Polymers, are commented directly onto the attached manuscript pdf file.

Overall, although many conclusions have been drawn by authors throughout the manuscript based on their experimental data, many assumptions have also been made based on literature analogues cases and not on the authors own findings eg page 8/16 lines 280-301. This is a major drawback of the manuscript that gives the impression of a review article and not a research based scientific work.

Author Response

Thank you for the valuable comments. Here, we attached the rebuttal for reviewer 3.

Round 2

Reviewer 1 Report

The authors answered adequately to my previous comments.

I consider the paper suitable to be published in Polymers in the present form.

Reviewer 2 Report

The authors did not respond to all the comments of the first revision, please send a detailed letter with the response of all the comments, it is the last chance otherwise the article will be rejected 

Author Response

Rebuttal for reviewer 2 is provided in the attachment. The manuscript has been re-edited and modified to minimize grammatical and typographical errors as much as possible.

Reviewer 3 Report

The authors have made convincing efforts to reply and revise the manuscript following  the comments of the reviewers.

The manuscript may now be accepted for publication.

Author Response

The rebuttal for reviewer 3 is stated in the attachment. The manuscript has been re-edited and modified to minimize grammatical and typographical errors as much as possible.

Round 3

Reviewer 2 Report

I see that the authors have now made the necessary revisions.
According to the English language, I leave that to the editorial offices.
The article is accepted for publication on my part